# Sexual Dysfunction of Patients with Diffuse Low-Grade Glioma: A Qualitative Review of a Neglected Concern

**DOI:** 10.3390/cancers14123025

**Published:** 2022-06-20

**Authors:** Arnaud Lombard, Hugues Duffau

**Affiliations:** 1Department of Neurosurgery, Centre Hospitalier Universitaire of Liège, 4000 Liège, Belgium; alombard@chuliege.be; 2Laboratory of Developmental Neurobiology, GIGA-Neurosciences, University of Liège, 4032 Liège, Belgium; 3Department of Neurosurgery, Gui de Chauliac Hospital, 34295 Montpellier, France; 4Team “Neuroplasticity, Stem Cells and Glial Tumors”, Institute of Functional Genomics, INSERM U-1191, University of Montpellier, 34090 Montpellier, France

**Keywords:** sexual activity, sexual dysfunction, low-grade glioma, surgery, chemotherapy, radiotherapy

## Abstract

**Simple Summary:**

Patients suffering from diffuse Low-Grade Glioma (LGG) are usually young adults and present long life expectancy thanks to multimodal therapeutic management. In this context, the preservation of quality of life (QoL) is essential, and sexual health is part of it. We reviewed here the current knowledge about sexual dysfunction in LGG patients. We highlighted how this issue has been largely neglected, despite an incidence from 44 to 62% in the rare series of the literature. Thus, there is a need to assess more systematically the occurrence of SD in clinical routine in order to adapt cancer treatments accordingly, to manage actively these troubles, and finally to improve patients’ QoL in the long run.

**Abstract:**

Diffuse low-grade gliomas (LGG) commonly affect young adults and display a slow evolution, with a life expectancy that can surpass 15 years, thanks to multimodal therapeutic management. Therefore, preservation of quality of life (QoL), including sexual health, is mandatory. We systematically searched available medical databases of Pubmed, Cochrane, and Scopus for studies that reported data on sexual activity or dysfunction (SD) in LGG patients. We analyzed results to determine incidence of SD and its association with QoL in this population. Three studies focused on SD incidence in patients presenting specifically LGG, or brain tumors including LGG. They comprised 124 brain tumor patients, including 62 LGG, with SD incidence ranging from 44 to 63%. SD was reported by more than 50% of interrogated women in the three studies. Regarding QoL, two out of the three studies found significant associations between SD and alterations of QoL parameters, particularly in the field of social and functional wellbeing. Finally, we discussed those results regarding methods of evaluation, inherent biases, and therapeutic implications regarding antiseizure medications and also planning of surgery, chemo-, and radiotherapy. Our review showed that SD is highly prevalent but still poorly studied in LGG patients. As those patients are usually young and enjoy an active life, there is a need to assess more systematically the occurrence of SD in clinical routine, in order to adapt cancer treatments accordingly, to manage actively these troubles, and finally to improve patients’ QoL in the long run.

## 1. Introduction

Gliomas account for half of all primary cerebral tumors and are classified as low- or high-grade [1]. Diffuse low-grade gliomas (LGG) display a distinct, slower evolution in comparison to high-grade gliomas (HGG). They usually affect young adults, who present no or few symptoms (usually seizures) at diagnosis [2]. As LGG unavoidably evolves to a higher grade of malignancy in absence of treatment, an early therapeutic approach, combining maximal safe resection(s), chemotherapy, and radiation therapy, is currently the gold standard [3]. This approach allows postponement of malignant transformation and hence improvement of overall survival to more than 15 years [4]. However, this therapeutic management should also be adapted to preserve quality of life (QoL), enabling these patients with long life expectancy to carry out normal private and professional activities. Sexual health is an essential component of this active life and has been shown to be associated with better health [5], and better health is associated with greater enjoyment of life [6]. Its disturbance significantly affects interpersonal relationships and overall QoL for both men and women [7]. Multiple studies have shown over the last few decades that 50 to 70% of cancer survivors, particularly from breast and gynecologic cancer, present a loss of desire [8,9]. Regarding LGG patients, Surbek et al. [10] first reported in 2015 an incidence of 44% from a cohort of 32 patients. In this context, we hypothesized that LGG patients presented an underreported, high incidence of SD, similar to that of cancer survivors. We first reviewed the actual knowledge of SD incidence and repercussions on QoL for LGG patients and then discussed methodology of evaluation as well as implications for surgical or medical treatments, addressing the different biases of the results. We finally proposed some perspectives on tailoring patient treatments by adapting to presented results.

## 2. Materials and Methods

### 2.1. Methods of Reviewing

We performed this systematic review in accordance with the Preferred Reporting Items for Systematic Reviews and Meta-Analyses (PRISMA) guidelines (registration number: 4hp6w) [11].

Criteria for considering studies for review: We considered clinical trials, prospective or retrospective cohort studies, case-control studies, and case series written in the English language for this review. Studies that considered sexual activity or dysfunction for adolescents and adult patients who present LGGs have been included. Studies with mixed tumor populations, such as series that included different types of brain tumors or different grades of gliomas, were considered as long as they included LGG patients.

Search methods for identification of studies and selection of studies: Major scientific databases, namely Medline by Pubmed, Scopus, CENTRAL by Cochrane, and ClinicalTrials.gov were searched from inception until April 2022. The search strategy used was [“sexual activity” OR “sexual disorder” OR “sexual dysfunction” OR “sexual sphere” OR “sexual health” OR “sexuality”] AND [“low grade glioma” OR “glioma” OR “LGG” OR “astrocytoma” OR “oligodendroglioma” OR “diffuse glioma” OR “brain tumor”] (see Appendix A). Handsearching of additional studies was done by going through the reference sections of included studies and relevant articles.

Data collection and analysis: Each study was sorted considering SD incidence or SD association with QoL. The following data were extracted from each study: title, citation, setting, design, total population of patients in the study, proportion of LGG if mixed, tumor location, methods of evaluation of SD, SD incidence, symptoms and onset, QoL parameters evaluated and their correlations with SD, timing of evaluation, and adjuvant treatments. When possible, mean and percentages were used to summarize study data. Then, methods of evaluation and therapeutic implications are discussed in a narrative manner.

### 2.2. Methods for Evaluation of Sexual Activity and SD

The most recent Diagnostic and Statistical Manual of Mental Disorders (DSM-5 2013) provides a classification system for healthcare professionals and describes seven main disorders for SD [12]. Four disorders are specific to males (i.e., delayed ejaculation, erectile disorder, hypoactive sexual desire disorder, and premature ejaculation), while three disorders apply only to females (i.e., female orgasmic disorder, female sexual interest/arousal disorder, and genito-pelvic pain/penetration disorder). Moreover, three diagnoses can apply to both sexes: substance/medication-induced SD, other specified SD, and unspecified SD.

Taking this into account, the first modality to address the modification of sexual activity is to determine the SD through an interview by one of the patient’s medical providers [13]. The second modality to evaluate SD in LGG patients consists of self-administered questionnaires. There are few well documented questionnaires, two to evaluate erectile dysfunction (the International Index of Erectile Function (IIEF) [14] and the Sexual Health Inventory for Men (SHIM) [15]), one questionnaire for the assessment of female sexual function (the Female Sexual Function Index (FSFI) [16,17]) and one applicable for both males and females (the Arizona Sexual Experience Scale (ASEX) [18]).

### 2.3. Methods for Evaluation of the Repercussions of SD on QoL

Several standardized and validated questionnaires allow evaluation of psychological status and QoL. The list here is not exhaustive but selects the essentials to understand presented results. The Hospital Anxiety and Depression Scale (HADS) [19] allows screening for clinically significant anxiety or depressive symptoms in ill patients, while the Psychological Distress Inventory (PDI) [20] allows measurement of psychological distress in cancer patients. Specific questionnaires were elaborated to evaluate patients’ QoL, such as EORTC QLQ-C30 [21] and EORTC QLQ-BN20 [22]. Moreover, the Functional Assessment of Cancer Therapy-Brain (FACT-Br) is another commonly-used instrument for measuring general QoL that reflects symptoms or problems associated with brain malignancies across five scales [23]. These scales include physical, social, emotional, and functional well-being, a subscale of specific brain cancer concerns, and a total well-being scale.

## 3. Results

### 3.1. Included Studies

We identified a total of three [10,24,25] studies, which address SD incidence and/or associations between SD and QoL (Figure 1). Handsearching of articles revealed 60 additional articles [1,2,3,4,5,6,7,8,9,11,12,13,14,15,16,17,18,19,20,21,22,23,26,27,28,29,30,31,32,33,34,35,36,37,38,39,40,41,42,43,44,45,46,47,48,49,50,51,52,53,54,55,56,57,58,59,60,61,62,63] needed for narrative purpose.

### 3.2. Incidence of SD and Consequences on QoL in LGG Adult Patients

In 2015, Surbeck et al. [10] reported the first study focusing on the sexual activity after surgery for LGG. Thirty-two patients, never irradiated and without treatment for at least a year following resection, completed anonymously a standardized questionnaire combining ASEX and a subjective statement. In this cohort, they reported a postoperative sexual change in 17 patients (53%), which was predominantly a deterioration (15 out of 17). Right-sided resections were associated with more difficulties in reaching orgasm, while temporal lobe resections in men were associated with reduction in sexual drive and sexual arousal. The most reported SD were related to sexual desire (34%), sexual arousal (34%), and orgasms (37.5%) (Table 1).

Using an ad hoc interview, Finocchiaro et al. [24] investigated the sexual activity of 46 patients with brain tumors, including 21 LGG patients. They observed that 58% of brain tumor patients reported SD at the time of evaluation and 34% reported that the onset of disturbance only occurs after diagnosis. Similar to Surbek et al. [9], the mainly-reported SD concerned sexual desire (56%), sexual arousal (41.5%), and orgasms (37.5%), with a trend to be more common in women. Finally, Boccia et al. [25] interrogated 46 brain tumor patients regarding sexual activity, including 9 LGG, using FSFI for 37 women and IIEF and erectile dysfunction scale for 13 men. In women, half of the patients met clinical criteria for dysfunction in the areas of arousal, lubrication, and satisfaction. Regarding the overall FSFI score, 67% met the criteria for dysfunction. There were also no significant effects of cancer type or left versus right hemisphere localization on FSFI scores. Only 11 male patients finally participated in the study, and 6 of them presented erectile dysfunction. The main reported SD symptoms concerned sex drive (62%) and satisfaction (62%) for men, arousal (46%) and satisfaction for women (52%).

As a whole, the presented studies report an incidence of SD that ranges from 44% to 63% in 124 brain tumor patients, including 62 LGG, and that appears to occur in more than 50% of interrogated women. It must be said that, in both studies from Finocchiaro et al. [24] and Boccia et al. [25], the specific incidence in LGG patients was not available.

Regarding QoL, Finocchiaro et al. [24] highlighted the fact that patients with sexual problems, detected by their ad hoc questionnaire, reported significantly higher levels of anxiety and depression as measured by HADS and significantly worse levels of QoL as measured by EORTC QLQ-C30 and EORTC QLQ-BN20 than patients who did not perceive adverse changes in their sexual sphere. They did not find any correlation between SD and distress evaluated with PDI. Using FSFI and FACT-Br to interrogate women, Boccia et al. [25] observed correlations between some FSFI and QoL subscales. Desire was correlated with functional wellbeing and total QoL scores, while social wellbeing was correlated with desire, arousal and satisfaction. In interrogated men, they did not observe a correlation between IIEF subscales or erectile dysfunction with the wellbeing subscales of the FACT-Br.

## 4. Discussion

### 4.1. Methods for Evaluation of Sexual Activity and SD in LGG Patients

As mentioned before, the first essential step to address the modification of sexual activity in LGG patients is to determine and to assess, as accurately as possible, the SD through an interview by one of the patient’s medical providers [13]. These interviews present several challenges. First of all, due to the patients’ understandable discomfort in disclosing their sexual activity, the interviewer has to create an appropriate climate, starting with basic questions and then allowing discussion of private sexual matters. In practice, however, it might be inconvenient to prolong some consultations where time is needed to discuss exam results or therapeutic options. Second, the interviewer should be prepared to encounter challenges in SD assessment due to bias, such as inconsistent definitions of sexual concepts, self-presentation bias, or cultural differences. In the specific case of LGG patients, the clinician should assess the context in which the SD occurs, such as sexual history or past or current relationships, and also how it could be linked to the disease. Indeed, it is essential to determine accurately the period when the symptoms appeared, according to the survivorship phase and/or treatment(s) [20], as SD might tie in with tumor evolution but also with adverse events due to surgery, adjuvant treatments, or medications (see below). Importantly, as LGG patients are of childbearing age, the desire for pregnancy and hence fertility preservation (FP) should also be discussed during this interview [13]. Given the difficulties in carrying out those interviews, it is not surprising that Laldjising et al. [26] reported that nearly 60% of interrogated neurosurgeons treating brain tumor patients never discussed sexual health with them, claiming that lack of knowledge and insufficient time were the main reasons. Discussing sexual health with tumor patients appears to also be challenging in clinical routine for oncologists and radiotherapists [27,28]. In Finocchiaro et al. [24], only seven patients (15.2%) received information regarding possible disorders in sexual activity by a family doctor or specialist.

The other modality for evaluating SD in LGG patients is the previously-cited self-administered questionnaires, which ease patients’ disclosure and also allow clinicians to track evolution over time [29]. The IIEF [14] allows for addressing erectile functioning, orgasmic functioning, sexual desire, intercourse satisfaction, and overall satisfaction. It is comprehensive, easily scored, and filled in less than 15 min, but it is less applicable to homosexual and bisexual men. The SHIM [15] is a widely-used screening measure in clinical practice but also in research to evaluate erectile dysfunction and can be used with heterosexual, homosexual, and bisexual men. The FSFI [16,17] is a 6- or 19-item scale for the assessment of female sexual function in five domains: satisfaction, desire, arousal (lubrication), orgasm, and pain/discomfort. This instrument is psychometrically valid and reliable for research use or for clinical practice and at diagnosis as well as during follow-up. Finally, the ASEX [18] is a five-item rating scale that quantifies sex drive, arousal, vaginal lubrication/penile erection, ability to reach orgasm, and satisfaction from orgasm. It enables quick, reliable evaluation for men and women [30]. The described questionnaires easily accommodate the consultation and can be filled before or after it.

Methods of evaluation of SD were different in the three presented studies. Surbek et al. used ASEX [10], Finocchiaro et al. [24] used an ad hoc questionnaire, and finally Boccia et al. [25] used FSFI for women and IIEF for men. This is an important bias that impedes accurate comparison between studies.

In the presented studies, QoL was evaluated in a distinct standardized questionnaire. However, evaluation of sexual activity could also be realized through a general questionnaire that could capture the full range of physical, mental, and social-health-related quality of life (HRQOL) issues. In a phase III study, including 24 gliomas, the recently developed and standardized QLQ-SURV111 questionnaire allows assessment of HRQOL in cancer survivors, in particular sexual concerns [31]. Anxiety and depression are frequently perceived by patients with brain tumors as soon as the diagnosis is announced and are both able to induce SD, particularly in terms of sexual desire [32,33]. Therefore, a questionnaire assessing both patient mood and sexual activity seems to be necessary. Ideally, a neurocognitive evaluation could be realized at the same time, eventually allowing correlation of SD with some cognitive impairment.

For clinical practice, we propose that an adequate questionnaire be filled by the patient, previously to the consultation, to more easily and accurately address SD occurrence at a dedicated time during the consultation. This evaluation should be realized systematically and longitudinally to detect a possible cause to the SD. Moreover, once SD is suspected, the medical provider should be prepared, if not trained, to help with resources or refer patients to a sexologist, gynecologist/urologist, or a specific FP unit [34]. It must be said that for select LGG patients, concurrent diseases might also participate in sexual activity disorder, in particular if one considers their prolonged survival. Endocrine troubles, as well as cardiovascular diseases, are classically implied in sexual dysfunction, and hence, specific patient care and evaluation might be needed.

### 4.2. Incidence of SD and Its Consequences on QoL in LGG Adult Patients: Limitations of Presented Results

In 2004, Krychman et al. [35] reported a first case of a 39-year-old patient suffering from a grade III glioma who benefited from specific care in a sexual health program, which in turn allowed her to resume sexual activity, emphasizing a putative need in brain tumor patients. More than ten years later, Surbeck et al. [10] reported the first study focusing on sexual activity after surgery for LGG. This study highlighted that SD were neglected in LGG patients but needed further evaluations, considering the size of the cohort and the lack of preoperative evaluation. Since then, only two studies have addressed SD incidence and emphasized SD association with QoL alterations. However, the studies of Finocchiaro et al. [24] and Boccia et al. [25] present important limitations. The studied population is heterogeneous, and researchers did not consider the patients’ stage of disease at the time of evaluation or the localization of the tumor, complicating evaluation of how their results specifically apply to LGG.

As mentioned in the introduction, multiple studies over the last few decades have already shown that breast, gynecologic, and urologic cancer survivors reported a high rate of loss of desire, ranging from 50 to 70% [8,9]. For those patients, interventional studies have already been carried out [36]. In addition to the limited number of studies available, this emphasize how poorly-studied and importantly neglected this concern is for LGG patients and how needed dedicated studies are in order to accurately evaluate SD incidence and its repercussions on QoL.

### 4.3. Antiseizure Medications and SD in LGG Patients

LGG patients usually present no or only mild symptoms at diagnosis. When LGG is not incidentally discovered, it is classically diagnosed after one or multiple seizures. Therefore, antiseizure medications (ASMs) are part of the daily LGG treatment and might be maintained for many years or even for life. The association between epilepsy and SD has been recently reviewed [37]. Different predisposing factors have been proposed, i.e., endocrine dysfunction, psychiatric comorbidities (depression and anxiety), psychosocial factors, and iatrogenic factors, particularly ASMs. On one hand, there is evidence to suggest that phenytoin, carbamazepine, and topiramate can induce SD in epileptic patients [38,39]. A few isolated case reports suggested that gabapentine, pregabaline, zonisamide, levetiracetam, and lacosamide can cause hyposexuality, erectile dysfunction, and anorgasmia [37]. On the other hand, a study of 141 patients on lamotrigine therapy suggested that lamotrigine can cause improvement in sexual functions [40]. In one patient presenting LGG-related epilepsy, erectile dysfunction disappeared after discontinuation of zonisamide [41]. Interestingly, Surbeck et al. [8] reported that remaining ASMs associated with right-sided resection in men were associated with higher overall ASEX scores than in women. Therefore, it is essential to consider patients’ ASMs when considering SD.

### 4.4. Surgical Treatment and SD in LGG Patients

As diffuse LGG is a brain disease in essence, tumor invasion within the parenchyma or cerebral damage related to treatment (e.g., surgical resection or irradiation) might impede the optimal functioning of neural structures involved in sexual activity. In the literature, multiple tumor cases with preoperative or postoperative SD have been reported [42]. Erickson described the first clearly-defined case of SD due to a cortical lesion. The patient experienced spontaneous sexual feelings in the vagina, caused by a tumor in the right paracentral lobule, which was suspected to irritate the area of cortical representation of the genitalia [43]. Then, while some authors reported an association between hypersexuality, epilepsy, and tumors in the temporal lobe (particularly the amygdala) [44], others observed an association between sexual impotence and tumor invasion of the limbic system (anterior hippocampus and cingulum) [42]. A few years later, Angelini et al. [45] described the case of a 10-year-old right-handed boy, admitted for seizures and behavioral disturbance (loss of social restraint, sexual polarization, aggressiveness), who presented a focal grade I glioma in the anterior right cingulum and who was free of symptoms after surgery. Suffering from an LGG in similar location (right cingulum and right frontal horn of the ventricle), a 21-year-old man presented with a decrease of libido and loss of ejaculation, associated with gonadotrophin deficiency: he completely recovered after surgery [46]. Another interesting case reported the occurrence of a postoperative association of mouthing of objects, abnormal sexual verbalization, loss of recognition of strangers, and fluent aphasia, compatible with a so-called Klüver–Bucy syndrome, after left anterior temporal lobectomy [47]. As a whole, the reported cases highlighted multiple structures, which could be implied in hypo- or hyper-sexuality, or both, thus unveiling hubs involved in the networks of sexual activity.

Moreover, in direct relation to sexual health, in case of desire for pregnancy in LGG women, several studies reported that there were significant rates of pregnancy-induced clinical decline or radiographic progression, with a higher risk of malignant transformation [48,49]. Interestingly, Ng et al. have recently demonstrated that the risk of glioma acceleration during pregnancy was correlated with postoperative tumor residue, i.e., with a significant longer survival after delivery if a complete resection of LGG was achieved before getting pregnant [50]. Taking this into account, if hormonal treatments are considered, a shorter time between follow-up might be indicated, avoiding missing tumor escape. These findings support the need not only to talk about sexual activity but also to advise LGG women with a desire for motherhood, which might open the discussion for (re)operation in order to reduce glioma volume prior to a possible pregnancy.

### 4.5. Chemotherapy, Radiotherapy and SD in LGG Patients

Chemotherapeutic agents usually used for LGG treatment are temozolomide or a combination of procarbazine, lomustine, and vincristine. Both treatments are related to loss of fertility, in turn related to their toxicity to gonads and to a suspected reduction of sex desire due to adverse events of drugs [51,52]. In the same way, brain irradiation is associated with a reduction of fertility due to a toxicity to the hypothalamo–hypophyseal axis [52]. While the consequences of chemo- and radiotherapy are well-established with regard to fertility and pregnancy, it is still not well-elucidated how those treatments specifically impact the sexual activity of LGG patients during or after the treatment.

In this context, prior to starting treatment, it is highly recommended to discuss with patients whether they have some desire to have a child. This is particularly relevant as LGG patients are of childbearing age and might benefit from FP. However, it appears that it is not that usual (Table 2). Indeed, Lehman et al. first reported, while interrogating 25 brain tumor survivors, that only 12 patients were aware of the risks of infertility secondary to cancer treatment [53]. Then, Stiner et al. showed in a study of 72 brain tumor female survivors, including 35 LGG, that only 30% of them had a discussion regarding FP prior to treatment [54]. In a cross-sectional survey of 1010 cancer survivors, Wide et al. also reported that patients with brain tumors are less likely to report being informed about potential impacts on their fertility and about FP [55]. On the other hand, in a population of 70 glioma patients with whom FP was discussed at diagnosis or progression, it appears that 51 patients (73%) accepted referral either to a sperm bank or a reproductive endocrinologist [56]. Importantly, patients undergoing FP prior to chemotherapy and/or radiation for a glioma achieve satisfactory FP outcomes in comparison to controls [57]. These results show that SD and FP should be discussed more systematically before any treatment, as well as during patient follow-up, in order to obtain a longitudinal evaluation.

### 4.6. Perspectives: Towards a Better Knowledge of Neural Bases of Sexual Activity to Tailor Treatments

In the past decade, advances in neurosciences, especially thanks to functional neuroimaging, have allowed investigation into the neural foundations underpinning sexual activity. This has led to the consideration of two entangled functionally different networks implied in sexual activity [58,59,60]. First, the “sexual wanting network” describes the sensory gateways to sexual motivation [59]. Briefly, using sexual images as visual inputs, the suggested network proposes that these sexual images are first processed in the activated occipitotemporal cortices in the same way as other visual inputs—particularly in the lateral part of the fusiform gyrus and the ventrolateral part of the inferior temporal gyrus—then conveyed to the amygdala and spread to the hypothalamus (for generation of autonomic responses), to the orbitofrontal cortex (OFC) and the anterior cingulate cortex (which are both reciprocally connected), to the nucleus accumbens, and to the anterior insula/claustrum. Second, the “sexual liking network” involves the inferior parietal lobule, hypothalamus, posterior insula, ventral premotor cortex, and the middle cingulate cortex [59]. Last, in women, the strongest orgasm-related changes in activity were seen in the mid-anterior portion of the left OFC and ventromedial prefrontal cortex, while failed orgasm attempts significantly enhanced the activity of the left lateral part of the OFC [61].

As the SD incidence in LGG has yet to be more clearly validated, it is premature to talk about adaptation of therapeutic strategy. However, as some specific structures in the networks described above (e.g., insula, amygdala, part of the OFC or uncus fasciculus) are often invaded by LGG [62] and hence targeted by surgical resection in order to achieve maximal resection [63], it would be interesting to determine how removal of those regions might impact patients’ sexual activity, especially in the right hemisphere [10]. Speculatively, we could hypothesize that alterations in some networks, preferentially in the right hemisphere that lead to behavior/emotion trouble [64], might increase the risk of SD occurrence in the context of mood spring. Adaptation of treatment strategy according to this improved knowledge and to the will of the patients remain questions to be addressed.

## 5. Conclusions

SD appears to be frequent in brain tumor patients and is still underreported. As an essential component of patient QoL, it is time to better assess SD occurrence in a more systematic way in clinical routine in LGG patients, with an adequate questionnaire associated with a dedicated time for discussion during consultation. Such an assessment should be performed longitudinally, before and after each treatment. Several factors should be considered (such as patient mental state, tumor evolution, antiepileptic treatment, surgical resection(s), chemo- and radiotherapy) for the possible onset of SD. Furthermore, specific resources must be provided in order to actively manage SD, especially by referring patients to a sexologist or to a dedicated FP unit. In sum, sexual health must be taken into consideration in LGG patients, who are usually young and enjoy an active life, in order to adapt their cancer treatments accordingly and ultimately to improve their QoL in the long run. Due to the small number of studies on SD and LGG in the current literature, our work is intended as a starting point to stimulate further research on this topic.

## Figures and Tables

**Figure 1 cancers-14-03025-f001:**
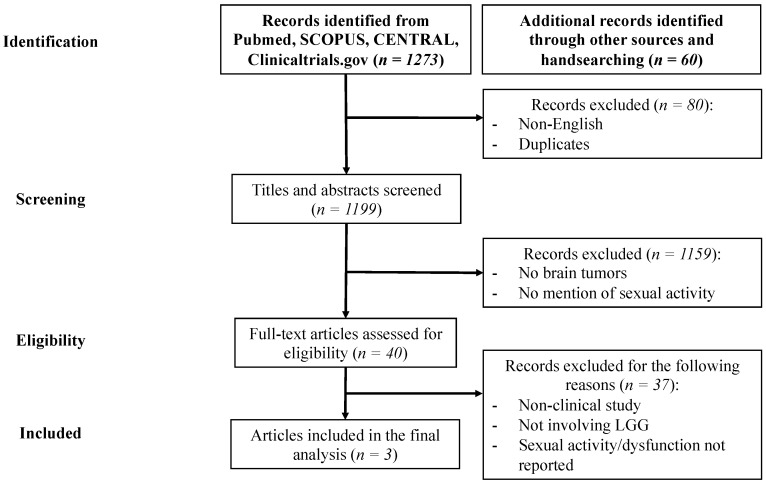
PRISMA flow diagram for study selection. From the 1273 articles identified in the four databases, 1199 were screened for titles and abstracts. Articles which did not concern brain tumors and/or sexual activity were excluded. From 40 articles assessed for eligibility, 37 were excluded due absence of LGG patients and/or report of sexual activity, or if the article was not a clinical study (e.g., case report). Finally, 3 studies were included for analysis.

**Table 1 cancers-14-03025-t001:** Incidence of SD and repercussions on QoL in studies considering brain tumors including LGG patients.

**First Author, Year**	Surbeck, 2015 [10]	Finocchiaro, 2017 [24]	Boccia, 2021 [25]
**Study type, country**	Transversal, France	Transversal, Italy	Transversal, USA
**Sample size (M/F) & Number of LGGs (M/F)**	32 (17/15), 32 (17/15)	46 (28/18), 21 (NR)	46 (11/35), 9 (3/6)
**Mean age (years)**	38.6	45	53.5
**Tumor Location** **(number)**	Frontal (11) Parietal (6) Occipital (2) Temporo-insular (6) Fronto-temporo-insular (7)	NR	Frontal (16) Temporal (8) Other locations (22)
**Tumor side** **(Number, %)**	Right (18, 56%) Left (14, 44%)	NR	NR
**Methods of SD evaluation**	ASEX	Ad hoc questionnaire	IIEF (M) and FSFI (F)
**SD general incidence (Number (M/F), % (M/F))**	14 (5/9), 44% (29%/60%)	27 (12/13), 58% (43%/72%)	29 (6/23), 63% (60%/67%)
**SD incidence in LGG (Number (M/F), % (M/F)**	14 (5/9), 44% (29%/60%)	NR	NR
**SD Symptoms** **(number, %)**	Sex drive (34%) Arousal (34%) Penile erection/vaginal lubrication (25%) Orgasm (37.5%) Satisfaction (28%)	Sex drive (56%) Arousal (41.5%) Pain (32%) Orgasm (37.5%) Satisfaction (32%)	Sex drive (M:62%, F:38%) Arousal (M: 54%, F:46%) Pain (M:NR, F:42%) Orgasm (M:54%, F:38%) Satisfaction (M:62%, F:50%)
**SD reported onset after diagnosis (%)**	47%	34%	NR
**Association** **between SD and tumor location**	- Right-sided resections and difficulties in reaching orgasm (*p* < 0.02) - Temporal lobe resection and reduction in sexual drive/arousal in men (*p* < 0.004) - ASMs, right-sided resection and higher ASEX scores in men (*p* = 0.031)	NR	- No association between tumor side and FSFI scores in women
**Methods of QoL evaluation**	NE	HADS, PDI, EORTC QLQ-C30, EORTC QLQ-BN20	FACT-Br
**Association between SD and QoL**	NE	- Significant higher levels of anxiety and depression (*p* < 0.01) and significant worse levels of QoL in patients with SD (*p* < 0.01) - No association between SD and distress	- In women, correlation of desire with functional wellbeing and total QoL scores, correlation of social wellbeing with desire, arousal and satisfaction. - In men, no correlation between SD and QoL
**Timing of** **Evaluation**	- At home - Return to normal social and professional life - No RT - No CT for at least 1 year - Not diagnosed or treated for depression	- During hospitalization or day treatments - No data about treatment timing	- Scheduled appointment - No data about treatment timing
**Adjuvant** **Treatments**	Surgery (32) RT (0) (No CT for at least 1 year)	NR	None (7) Surgery (8) Surgery + CT (1) Surgery + RT (2) All 3 (24) Missing data (8)

M—Male; F—Female; ASMs—Antiseizure medications; CT—Chemotherapy; RT—Radiotherapy; NR—Not Reported; NE—Not Evaluated.

**Table 2 cancers-14-03025-t002:** Studies addressing FP discussion and results in LGG patients.

**First Author, Year**	Lehman, 2018 [53]	Stiner, 2019 [54]	Wide, 2021 [55]	Stone, 2017 [56]	Nordan, 2020 [57]
**Number of Cancer Patients (M/F)**	92 (NR)	69 (0/69)	1010 (316/694)	70 (38/32)	10 (0/10)
**Number of Brain Tumors (LGG)**	25 (NR)	69 (35)	123 (NR)	70 (12)	10 (5)
**Number of Brain Tumor Patients that Had FP Discussion Prior to Treatment**	12	21	23	70 (47 at diagnosis, 23 at progression)	10
**% of Brain Tumor Patients that Had FP Discussion Prior to Treatment**	50%	30%	19%	100% (67% at diagnosis, 33% at progression)	100%
**Other results**		- While fertility preservation was not important at the time of diagnosis, it was a priority for them at the time of survey completion		- 51 patients (73%) accepted referral to specific unit - Patients were more likely to accept referral if they had no prior children	- The total number of follicles, of oocytes retrieved and the percentage of mature oocytes were similar between cases and controls. - One patient delivered a healthy child.

M—Male; F—Female; NR—Not Reported.

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
