# Peer review of "Sexual Dysfunction of Patients with Diffuse Low-Grade Glioma: A Qualitative Review of a Neglected Concern"

_cancers, 2022, doi:10.3390/cancers14123025_

Round 1
Reviewer 1 Report
The manuscript is well written and the topic is of interest and often neglected by the neuro-oncological literature.
Authors performed a systematic review on the sexual dysfunction (SD) of LGG patients. They did so by analyzing the methods of evaluation and the incidence of SD. Authors reported that they included in the final sample 16 articles, including a total of 124 patients. Results showed an incidence of SD between 44% to 62% of the evaluated cases (more in females) In two studies included in the review, there was a significant correlation between sexual activity and QoL.
Authors conclude that, since LGG involves young people, patients' sexual activity should be taken into consideration more often in the clinical routine in order to adapt cancer treatments, as it is an index contributing to their quality of life.
I have the following observations to the authors attention:
Introduction section:
-The introduction section could be more focused on data in literature showing how in the clinical population SD could impact patients' quality of life.
-there are no hypothesis put forward by the authors
Method section:
-there is a confusion on the number of studies included: the abstract reports 16 studies while in the results and in the Figure 1 flow-chart it appears that the number of studies was 11.
there is a very long section in the Results reporting the methods that can be used in order to evaluate SD in LGG patients. The reader expects to find information about how did the authors of the studies included in the present meta-analyses the patients' SD of their own studies. By contrast, here, the reader finds a general paragraph focused on how do the authors of the present paper would test SD in LGG patients.
-There is also a proposal on the use of an adequate questionnaire they would use (this could be part of the discussion section under the paragraph future studies.
-It is unclear why the studies cited in "Surgical treatment and SD in LGG patients" do not appear in Table 1
In the paragraph "Perspectives: towards a better knowledge of neural bases of sexual activity to tailor 284
treatments" authors should pay attention to the attribution of fMRI activations to specific processing. They mention that "sexual images are first processed in the activated occipito-temporal cortices, especially the lateral part of the fusiform gyrus and the ventrolateral part of the inferior temporal gyrus [...]"
These areas are activated not because subjects are processing sexual stimuli, but because these areas (e.g., extrastriate body area" selectively respond to the view of body (full bodies, body parts") thus as it is reported there is a mistake as it seems that these areas are related to the processing of sexual stimuli.
Author Response
The manuscript is well written and the topic is of interest and often neglected by the neuro-oncological literature.
Authors performed a systematic review on the sexual dysfunction (SD) of LGG patients. They did so by analyzing the methods of evaluation and the incidence of SD. Authors reported that they included in the final sample 16 articles, including a total of 124 patients. Results showed an incidence of SD between 44% to 62% of the evaluated cases (more in females) In two studies included in the review, there was a significant correlation between sexual activity and QoL.
Authors conclude that, since LGG involves young people, patients' sexual activity should be taken into consideration more often in the clinical routine in order to adapt cancer treatments, as it is an index contributing to their quality of life.
I have the following observations to the authors attention:
Introduction section:
-The introduction section could be more focused on data in literature showing how in the clinical population SD could impact patients' quality of life.
-there are no hypothesis put forward by the authors
Author’s response: We thank the Reviewer 1 for this relevant observation. Introduction has been modified accordingly.
Method section:
-there is a confusion on the number of studies included: the abstract reports 16 studies while in the results and in the Figure 1 flow-chart it appears that the number of studies was 11.
Author’s response: This has been corrected and adapted to modifications of the manuscript.
-there is a very long section in the Results reporting the methods that can be used in order to evaluate SD in LGG patients. The reader expects to find information about how did the authors of the studies included in the present meta-analyses the patients' SD of their own studies. By contrast, here, the reader finds a general paragraph focused on how do the authors of the present paper would test SD in LGG patients.
Author’s response: We thank Reviewer 1 for this observation. We adapted the manuscript in order to describe general methods of evaluation of SD and QoL in the Methods section and to discuss the methods used in reported studies in the Discussion section.
-There is also a proposal on the use of an adequate questionnaire they would use (this could be part of the discussion section under the paragraph future studies.
Author’s response: We adapted the manuscript in order to propose the most adequate questionnaire in the Discussion section (line 223 to 233).
-It is unclear why the studies cited in "Surgical treatment and SD in LGG patients" do not appear in Table 1
Author’s response: As Table 1 focuses on incidence of SD and its repercussions on QoL, the reported cases in the "Surgical treatment and SD in LGG patients" section do not bring data that can be compared with the other studies. We adapted and completed Table 1 to compare more easily the different studies.
In the paragraph "Perspectives: towards a better knowledge of neural bases of sexual activity to tailor treatments" authors should pay attention to the attribution of fMRI activations to specific processing. They mention that "sexual images are first processed in the activated occipito-temporal cortices, especially the lateral part of the fusiform gyrus and the ventrolateral part of the inferior temporal gyrus [...]"
These areas are activated not because subjects are processing sexual stimuli, but because these areas (e.g., extrastriate body area" selectively respond to the view of body (full bodies, body parts") thus as it is reported there is a mistake as it seems that these areas are related to the processing of sexual stimuli.
Author’s response: We corrected the manuscript by mentioning that the processing in the occipito-temporal cortices was not specific to sexual stimuli.
Reviewer 2 Report
The review is very interesting, because it covers a still unexplored filed in the care and management of patients affected by primary brain tumors.
It is well written and it provides useful informations that can be utilized in clinical practice.
I suggest to change "anti epileptic drugs (AEDs)" with "anti seizure medications (ASMs)", according to the new ILAE recommendations"
Furthermore, I suggest to change the word "gay" with "male homosexuals"
Author Response
The review is very interesting, because it covers a still unexplored filed in the care and management of patients affected by primary brain tumors.
It is well written and it provides useful informations that can be utilized in clinical practice.
I suggest to change "anti epileptic drugs (AEDs)" with "anti seizure medications (ASMs)", according to the new ILAE recommendations"
Author’s response: We thank Reviewer 2 for this observation. We corrected the manuscript accordingly.
Furthermore, I suggest to change the word "gay" with "male homosexuals"
Author’s response: We corrected the manuscript accordingly.
Reviewer 3 Report
Excellent narrative review on an under-recognised topic. This is a rare case where I have no suggestions beyond congratulating the authors for their excellent work!
Author Response
Excellent narrative review on an under-recognised topic. This is a rare case where I have no suggestions beyond congratulating the authors for their excellent work!
Author’s response: We thank the Reviewer for his/her positive comments
Reviewer 4 Report
Title: “Sexual dysfunction of patients with diffuse low grade glioma: a systematic review of a neglected concern“
In this work the authors performed a review regarding the sexual dysfunction (SD) of LGG patients and the implications for treatments. They used available medical databases of Pubmed, Cochrane and Scopus for studies that reported data on sexual activity in LGG patients. They analyzed methods of evaluation and incidence of SD in this population, as well as therapeutic implications. A total of 16 studies or reported cases were included in the review, in particular, four studies were related to methods of evaluation, while three studies and one case report focused on SD incidence in patients presenting LGG. They comprised 124 patients, with SD incidence ranging from 44% to 62 %. SD were reported by more than 50% of interrogated women in the three studies. The authors underlined that two of them found significant correlation between sexual activity and QoL.
General comment: Although the claims of the authors, this work seems to be not able to support in a clear (and quantitative) way the correlation between SD and LGG. In addition, from the main text is not clear whether the chosen cohort of patients is statistically representative for a more wide population. Unfortunately, it seems that a clear approach based on statistical inference is lacking in this work. In addition, a large part of the claims is based on interviews, which, as underlined by the authors, “… present several challenges”. This challenges are partially detailed in the main text. As a consequence, the whole reliability and then the value of this work, is definitively not clear. Moreover, the standard sections of a scientific contribution are mixed in the current version of the main text. Therefore, also the formal presentation of the work should be deeply reworked and improved. This work can not be published.
Some detailed comment:
2.1. Criteria for considering studies for review 5
2.2. Search methods for identification of studies and selection of studies
2.3. Data collection and analysis
*) These sections seem not to explain in a clear way the statistical methods used by authors along their main text. Please rework and improve.
3. Results
*) This section provides the results of the search not the results of the analysis performed by these (or others) authors on the literature, which have to be statistically significant and should show that SD and LGG are (at least) positively correlated. Some medical reasons for a causation relationship, could be (and should be) also provided (and discussed) to further support the only “positive” correlation.
Lines:"3.2. Methods for evaluation of sexual activity and SD in LGG patients
These interviews present several challenges. First of all, due to the understandable patients’ discomfort to disclose their sexual 101
activity, the interviewer has to create an appropriate climate, starting with basic questions 102
and then allowing to discuss private sexual matters. In practice, it might be however dif- 103
ficult to prolong some consultations where time is needed to discuss exam results or ther- 104
apeutic options. Second, the interviewer should be prepared to encounter challenges in 105
SD assessment due to bias, such as inconsistent definitions of sexual concepts, self-presen- 106
tation bias or cultural differences. In the specific case of LGG patients, the clinician should 107
assess the context in which the SD occur, such as sexual history, past or current relation- 108
ships, and also how it could be linked to the disease. Indeed, it is essential to determine 109
accurately the period when the symptoms appeared, according to the survivorship phase
and/or treatment(s)20, as SD might tie in with tumor evolution but also with adverse events 111
due to surgery, adjuvant treatments or medications (see below). Importantly, as LGG pa- 112
tients are of childbearing age, the desire of pregnancy and hence fertility preservation (FP) 113
should also be discussed during this interview19. Given the difficulties in carrying out 114
those interviews, it is not surprising that Laldjisng et al.21 reported that nearly 60% of in- 115
terrogated neurosurgeons treating brain tumor patients never discussed sexual health 116
with them, claiming that lack of knowledge and insufficient time were the main reasons. 117
Discussing sexual health with tumor patients appears to be also challenging in clinical 118
routine for oncologists and radiotherapists22,23
Another modality to evaluate SD in LGG patients consists of self-administered ques- 120
tionnaires, which ease patients’ disclosure and also allow clinicians to track evolution over 121
time24. There are few well documented questionnaires, two to evaluate erectile dysfunc- 122
tion and one questionnaire for the assessment of female sexual function. The International 123
Index of Erectile Function (IIEF)25 allows to address erectile functioning, orgasmic func- 124
tioning, sexual desire, intercourse satisfaction, and overall satisfaction. It is comprehen- 125
sive, easily scored, filled in less than 15 minutes but is less applicable to gay and bisexual 126
men. The Sexual Health Inventory for Men (SHIM)26 is a widely used screening measure 127
in clinical practice but also in research to evaluate erectile dysfunction and can be used 128
with heterosexual, gay, and bisexual men. The Female Sexual Function Index27,28 is a 6- or 129
19-item scale for the assessment of female sexual function in five domains: satisfaction, 130
desire, arousal (lubrication), orgasm, and pain/discomfort. This instrument is psychomet- 131
rically valid and reliable for research use or for clinical practice, at diagnosis as well as 132
during follow-up. In addition to those three questionnaires, the Arizona Sexual Experi- 133
ence Scale29 is a 5-item rating scale that quantifies sex drive, arousal, vaginal lubrica- 134
tion/penile erection, ability to reach orgasm, and satisfaction from orgasm. It enables 135
quick, reliable evaluation for men and women30. The described questionnaires accommo- 136
date easily the consultation and can be filled before or after it. 137
Evaluation of sexual activity could also be realized through a general questionnaire 138
that could capture the full range of physical, mental and social health-related quality of 139
life (HRQOL) issues. In a phase III study, including 24 gliomas, the recently developed 140
and standardized QLQ-SURV111 questionnaire allowed to assess HRQOL in cancer sur- 141
vivors, in particular sexual concerns7. Anxiety and depression are frequently perceived by 142
patients with brain tumors, as soon as the diagnosis is announced, and they are both able 143
to induce SD, particularly trouble of sexual desire31,32. Therefore, a questionnaire assessing 144
both patient’s mood and sexual activity seems to be necessary. Ideally, a neurocognitive 145
evaluation could be performed in the same time, eventually allowing to correlate SD with 146
some cognitive impairment. 147
For clinical routine, we propose that an adequate questionnaire would be filled by 148
the patient, previously to the consultation, allowing to address more easily and accurately 149
the SD occurrence at a dedicated time during the consultation. This evaluation should be 150
realized systematically and longitudinally to detect a possible cause to the SD. Moreover, 151
the medical provider should be prepared, nay trained, to help with resources, especially 152
being able to address patients to a sexologist or to specific FP unit33. It has to be said that 153
for selected LGG patients, concurrent diseases might also participate to sexual activity 154
disorder, in particular if one considers their prolonged survival. Endocrine troubles, as 155
well as cardiovascular diseases, are classically implied in SD and hence specific patient 156
care and evaluation might be needed."
*) This paragraph is related to the methodology (see the title) and should be moved in the "Methods" section.
lines: "Given the difficulties in carrying out 114
those interviews, it is not surprising that Laldjisng et al.21 reported that nearly 60% of interrogated neurosurgeons treating brain tumor patients never discussed sexual health 116
with them, claiming that lack of knowledge and insufficient time were the main reasons. 117
Discussing sexual health with tumor patients appears to be also challenging in clinical 118
routine for oncologists and radiotherapists22,23. 11"
*) These lines explicitly underline that interviews are challenging, thus their value seem to be not clear to support the results claimed in this work.
3.3. Incidence of SD in LGG adult patients and consequences on QoL (Table 1). Table 1. Incidence of SD in LGG patients reported in studies
*) This paragraph and this table should be one of the main results of this work.
However, the caption of the table is too short and readers should refer to the text to understand the meaning of the table. In addition, it seems that, again, no statistically significant procedures are reported to support all the claims of this work. This part should be enlarged, deeply reworked and improved providing the needed quantitative analysis.
3.4. Antiepileptic treatment and SD in LGG patients
3.5. Surgical treatment and SD in LGG patients
3.6. Chemotherapy, radiotherapy and SD in LGG patients
*) Again for all these paragraphs the results are not presented in a clear way. In particular, the statistical relevance of the claims is not presented and a mix between a "Discussion" section and a "Results" section is provided. As a consequence, the authors should deeply rework this section to improve it.
Table 2. Studies addressing FP discussion and results in LGG patients.
*) See the previous comments. The table and the caption of this table should be more clear and the text of the caption enlarged in a better description.
4. Conclusion
*) Conclusions appear to be not quantitatively supported by the main text of this manuscript.
Author Response
In this work the authors performed a review regarding the sexual dysfunction (SD) of LGG patients and the implications for treatments. They used available medical databases of Pubmed, Cochrane and Scopus for studies that reported data on sexual activity in LGG patients. They analyzed methods of evaluation and incidence of SD in this population, as well as therapeutic implications. A total of 16 studies or reported cases were included in the review, in particular, four studies were related to methods of evaluation, while three studies and one case report focused on SD incidence in patients presenting LGG. They comprised 124 patients, with SD incidence ranging from 44% to 62 %. SD were reported by more than 50% of interrogated women in the three studies. The authors underlined that two of them found significant correlation between sexual activity and QoL.
General comment: Although the claims of the authors, this work seems to be not able to support in a clear (and quantitative) way the correlation between SD and LGG. In addition, from the main text is not clear whether the chosen cohort of patients is statistically representative for a more wide population. Unfortunately, it seems that a clear approach based on statistical inference is lacking in this work. In addition, a large part of the claims is based on interviews, which, as underlined by the authors, “… present several challenges”. This challenges are partially detailed in the main text. As a consequence, the whole reliability and then the value of this work, is definitively not clear. Moreover, the standard sections of a scientific contribution are mixed in the current version of the main text. Therefore, also the formal presentation of the work should be deeply reworked and improved. This work can not be published.
Author’s response: We understand Reviewer 4’s concerns. In this context, we changed the manuscript in order to present the methodology of reviewing, of SD evaluation and of QoL evaluation in the Methods section. We only selected studies that specifically focus on SD incidence and its repercussions on QoL in the Results section and we completed the Table 1 accordingly. Finally, even though we performed this systematic review in accordance with the PRISMA guidelines (OSF registration number: 4hp6w), we discussed narratively the different biases and limitations of the study, as well as therapeutic implications and perspectives.
Some detailed comment:
2.1. Criteria for considering studies for review 5
2.2. Search methods for identification of studies and selection of studies
2.3. Data collection and analysis
*) These sections seem not to explain in a clear way the statistical methods used by authors along their main text. Please rework and improve.
Author’s response: This section was adapted according to the changes described above. However, due to the restricted number of studies, further statistical analyses seem not to be suitable to the acquired data.
- Results
*) This section provides the results of the search not the results of the analysis performed by these (or others) authors on the literature, which have to be statistically significant and should show that SD and LGG are (at least) positively correlated. Some medical reasons for a causation relationship, could be (and should be) also provided (and discussed) to further support the only “positive” correlation.
Author’s response: Even if we understand Reviewer 4’s comment, we are not able to realize statistical analyses based upon the presented data, due to the restricted numbers, but also due to the discussed biases in the studies addressing SD.
Lines:"3.2. Methods for evaluation of sexual activity and SD in LGG patients
These interviews present several challenges. First of all, due to the understandable patients’ discomfort to disclose their sexual activity, the interviewer has to create an appropriate climate, starting with basic questions and then allowing to discuss private sexual matters. In practice, it might be however difficult to prolong some consultations where time is needed to discuss exam results or therapeutic options. Second, the interviewer should be prepared to encounter challenges in SD assessment due to bias, such as inconsistent definitions of sexual concepts, self-presentation bias or cultural differences. In the specific case of LGG patients, the clinician should assess the context in which the SD occur, such as sexual history, past or current relationships, and also how it could be linked to the disease. Indeed, it is essential to determine accurately the period when the symptoms appeared, according to the survivorship phase and/or treatment(s)20, as SD might tie in with tumor evolution but also with adverse events due to surgery, adjuvant treatments or medications (see below). Importantly, as LGG patients are of childbearing age, the desire of pregnancy and hence fertility preservation (FP) should also be discussed during this interview. Given the difficulties in carrying out those interviews, it is not surprising that Laldjisng et al.21 reported that nearly 60% of interrogated neurosurgeons treating brain tumor patients never discussed sexual health with them, claiming that lack of knowledge and insufficient time were the main reasons. Discussing sexual health with tumor patients appears to be also challenging in clinical routine for oncologists and radiotherapists22,23
*) This paragraph is related to the methodology (see the title) and should be moved in the "Methods" section.
Author’s response: We thank Reviewer 4 for this observation. We adapted the manuscript accordingly.
lines: "Given the difficulties in carrying out those interviews, it is not surprising that Laldjisng et al.21 reported that nearly 60% of interrogated neurosurgeons treating brain tumor patients never discussed sexual health with them, claiming that lack of knowledge and insufficient time were the main reasons. Discussing sexual health with tumor patients appears to be also challenging in clinical routine for oncologists and radiotherapists22,23. 11"
*) These lines explicitly underline that interviews are challenging, thus their value seem to be not clear to support the results claimed in this work.
Author’s response: As described above, this review aims to highlight a neglected concern and to reveal the poor actual knowledge in a narrative way.
3.3. Incidence of SD in LGG adult patients and consequences on QoL (Table 1). Table 1. Incidence of SD in LGG patients reported in studies
*) This paragraph and this table should be one of the main results of this work.
However, the caption of the table is too short and readers should refer to the text to understand the meaning of the table. In addition, it seems that, again, no statistically significant procedures are reported to support all the claims of this work. This part should be enlarged, deeply reworked and improved providing the needed quantitative analysis.
Author’s response: We thank Reviewer 4 for this observation. We reworked Table 1 accordingly. Still, no statistical analyses were realized, as discussed earlier.
3.4. Antiepileptic treatment and SD in LGG patients
3.5. Surgical treatment and SD in LGG patients
3.6. Chemotherapy, radiotherapy and SD in LGG patients
*) Again for all these paragraphs the results are not presented in a clear way. In particular, the statistical relevance of the claims is not presented and a mix between a "Discussion" section and a "Results" section is provided. As a consequence, the authors should deeply rework this section to improve it.
Author’s response: We thank Reviewer 4 for this observation. We adapted the manuscript and discussed those sections in the Discussion section.
Table 2. Studies addressing FP discussion and results in LGG patients.
*) See the previous comments. The table and the caption of this table should be more clear and the text of the caption enlarged in a better description.
Author’s response: We thank Reviewer 4 for this observation. We adapted the Table 2 accordingly.
- Conclusion
*) Conclusions appear to be not quantitatively supported by the main text of this manuscript.
Author’s response: We adapted the manuscript to highlight the important results, even if the actual knowledge in the field is highly limited.
Reviewer 5 Report
The authors have made a well-stated systematic review to express the concern of sexual dysfunction (SD) of LGG patients, and highlight the implications for treatments. Several detailed problems remained to be resolved to improve the manuscript.
- Please mention some key implications for treatments in the Abstract.
- I believe the methods for evaluation of sexual activity and SD in LGG patients would be different in patients who have not yet received or already received surgery. Is there any literature describing it or what are the authors’ opinions?
- Findings from Surbeck et al. (line 168) indicated the relationship between surgical locations and SD, which seemed very interesting. Is there any physiological explanation or implication for surgery?
- Psychological treatments or medications might be suggested for SD patients. Please discuss if there are some differences or attention points when applying these methods between general SD patients or LGG SD patients.
Author Response
The authors have made a well-stated systematic review to express the concern of sexual dysfunction (SD) of LGG patients, and highlight the implications for treatments. Several detailed problems remained to be resolved to improve the manuscript.
Please mention some key implications for treatments in the Abstract.
Author’s response: We thank Reviewer 5 for this observation. We re-phrased the abstract accordingly.
I believe the methods for evaluation of sexual activity and SD in LGG patients would be different in patients who have not yet received or already received surgery. Is there any literature describing it or what are the authors’ opinions?
Author’s response: As reported in the "Surgical treatment and SD in LGG patients" section, LGG patients presented very rarely with SD. We did not find studies that evaluate preoperatively the sexuality of LGG patients. However, in the studies of Surbek et al. and Finocchiaro et al., 15 out of 32 patients and 27 out of 46, respectively, reported a postoperative deterioration of their sexual activity. As mentioned in the Discussion and Conclusion, we believe that such an evaluation should be performed more systematically and longitudinally (before and after each treatment) in order to determine exactly how surgery impacts sexual activity, and that the method of assessment should be standardized and identical before and after surgery. Nonetheless, preoperative evaluation presents several challenges, considering patient’s state of mind at the difficult period of diagnosis.
Findings from Surbeck et al. (line 168) indicated the relationship between surgical locations and SD, which seemed very interesting. Is there any physiological explanation or implication for surgery?
Author’s response: It seems currently premature to determine some surgical strategies based on the results reported by Surbek et al. We could hypothesize that alterations in some networks, preferentially in the right hemisphere and leading to behavior/emotion troubles (Lemaitre et al. 20211), might increase the risk of SD occurrence, in the context of mood spring. However, we believe that this is too speculative.
Psychological treatments or medications might be suggested for SD patients. Please discuss if there are some differences or attention points when applying these methods between general SD patients or LGG SD patients.
Author’s response: LGG patients are usually young and present no or little comorbities. In case of SD occurrence, as long as the SD diagnosis is achieved, all patients might benefit from adapted treatments, without specific restrictions. Once SD is diagnosed, medical provider should propose to the patient to meet a sexologist and, only if needed following his/her evaluation, should consider to address the patient to a urologist or gynecologist. SD covers a large spectrum of troubles and treatments are adapted accordingly. The neuro-oncological medical provider seems to be the right person to identify SD but might not treat it alone, unless he is trained for it. Moreover, as discussed in the manuscript, every patient that will need chemo- or radiotherapy should have heard about fertility preservation.
Bibliography
- Lemaitre A-L, Herbet G, Duffau H, Lafargue G. Personality and behavioral changes after brain tumor resection: a lesion mapping study. Acta Neurochir (Wien). 2021;163(5):1257-1267.
Round 2
Reviewer 4 Report
Title: “Sexual dysfunction of patients with diffuse low grade glioma: a systematic review of a neglected concern “
In this work the authors performed a review regarding the sexual dysfunction (SD) of LGG patients and the implications for treatments. They used available medical databases of Pubmed, Cochrane and Scopus for studies that reported data on sexual activity in LGG patients. They analyzed methods of evaluation and incidence of SD in this population, as well as therapeutic implications. A total of 16 studies or reported cases were included in the review, in particular, four studies were related to methods of evaluation, while three studies and one case report focused on SD incidence in patients presenting LGG. They comprised 124 patients, with SD incidence ranging from 44% to 62 %. SD were reported by more than 50% of interrogated women in the three studies. The authors underlined that two of them found significant correlation between sexual activity and QoL.
General comment: Although the authors partially reviewed their manuscript, the main concern about the value of this work still remain. Indeed, it is not clear, from a statistical and quantitative point of view, whether a correlation between SD and LGG really exists. I appreciate the words of the authors which describe that:
“As mentioned before, the first essential step to address the modification of sexual 179
activity in LGG patients is to determine and to assess, as accurately as possible, the SD 180
through an interview by one of the patient’s medical providers [13]. These interviews pre- 181
sent several challenges. First of all, due to the understandable patients’ discomfort to dis- 182
close their sexual activity, the interviewer has to create an appropriate climate, starting 183
with basic questions and then allowing to discuss private sexual matters. “
and again:
“Methods of evaluation of SD were different in the three presented studies. Surbek et 218
al. used ASEX [10], Finocchiaro et al. [24] used an ad hoc questionnaire and finally Boccia
et al. [25] used FSFI for women and IIEF for men. This is an important bias and impede 220
the accurate comparison between studies. 221”
and furthermore:
“It has to be said that for selected LGG patients, concurrent diseases 239
might also participate to sexual activity disorder, in particular if one considers their pro- 240
longed survival. Endocrine troubles, as well as cardiovascular diseases, are classically im- 241
plied in sexual dysfunction and hence specific patient care and evaluation might be 242
needed. “
Thus, the authors describe as the crucial part of the work is to collect data through interviews, which are highly challenging. In addition, different authors used different methods to evaluate the effectiveness of their work, thus an accurate comparison between different studies seems to be lacking. Finally, the authors admits that “concurrent diseases” i.e., endocrine troubles, cardiovascular diseases, “are classically implied in sexual dysfunction”. Therefore, a specific patient care and evaluation might be needed
In conclusion, this work is still not able to quantitatively correlate SD and LGG, as already pointed out in the previous version. This work can not be published at this stage and several changes should be provided to improve the quality of this work. Please refer to the commentsof the previous round of revisions to improve the quality of the work.
Author Response
We thank the reviewer for this comment. The manuscript was adapted accordingly (line 324-326, line 386-389).